# Transient Improvement after Switch to Low Doses of RimabotulinumtoxinB in Patients Resistant to AbobotulinumtoxinA

**DOI:** 10.3390/toxins12110677

**Published:** 2020-10-27

**Authors:** Harald Hefter, Sara Samadzadeh, Marek Moll

**Affiliations:** Department of Neurology, University of Düsseldorf, Moorenstraße 5, D-40225 Düsseldorf, Germany; sara.samadzadeh@yahoo.com (S.S.); marek.moll@med.uni-duesseldorf.de (M.M.)

**Keywords:** secondary non-response, antibody induction, botulinum toxin type A, botulinum toxin type B, cervical dystonia

## Abstract

Botulinum toxin type B (BoNT/B) has been recommended as an alternative for patients who have become resistant to botulinum toxin type A (BoNT/A). This study aimed to compare the clinical effect, within a patient, of four injections with low doses of rimabotulinumtoxinB with the effect of the preceding abobotulinumtoxinA (aboBoNT/A) injections. In 17 patients with cervical dystonia (CD) who had become resistant to aboBoNT/A, the clinical effect of the first four rimabotulinumtoxinB (rimaBoNT/B) injections was compared to the effect of the first four aboBoNT/A injections using a global assessment scale and the TSUI score. After the first two BoNT/B injections, all 17 patients responded well and to a similar extent as to the first two BoNT/A injections, but with more side effects such as dry mouth and constipation. After the next BoNT/B injection, the improvement started to decline. The response to the fourth BoNT/B injection was significant (*p* < 0.048) lower than the fourth BoNT/A injection. Only three patients developed a complete secondary treatment failure (CSTF) and five patients a partial secondary treatment failure (PSTF) after four BoNT/B injections. In nine patients, the usual response persisted. With the use of low rimaBoNT/B doses, the induction of CSTF and PSTF to BoNT/B could not be avoided but was delayed in comparison to the use of higher doses. In contrast to aboBoNT/A injections, PSTF and CSTF occurred much earlier, although low doses of rimaBoNT/B had been applied.

## 1. Introduction

Since the first clinical application of botulinum neurotoxin type A (BoNT/A) to correct extraocular muscle dysbalance as an alternative to strabismus surgery by Scott in 1979 [1], injections of BoNT/A have become the treatment of choice for a variety of focal dystonia [2,3]. To maintain a significant level of improvement, repetitive injections have to be performed. Soon after a more frequent clinical use of BoNT/A, it was realized that between 4.3 and 17% of the BoNT/A-treated patients may develop neutralizing antibodies (NABs) [4,5] or an immunoresistance with the clinical implication of a partial or complete secondary treatment failure (PSTF or STF) in up to 40% of patients [6]. In 1997 the new onabotulinumtoxinA preparation (onaBoNT/A; Botox^®^; Allergan) became available with a reduction of the protein content by a factor 5–6 to 5 ng/vial per 100 MU (summary of product characteristics (SPC) revised: 7/2020) and a neurotoxin protein load of 0.73 ng per 100 U [7]. This preparation also has a more than five to six-fold lower risk of antibody formation of about 1.2% [5]. But even for this new onaBoNT/A preparation, the incidence of NABs is not zero. For the abobotulinumtoxinA preparation (aboBoNT/A; Dysport^®^; Ipsen) with a comparable neurotoxin protein load of 0.65 ng/100 U [7] (which was licensed 1990 in Europe) a similar or probably higher risk for antibody formation has to be expected since potency units are not the same for aboBoNT/A and onaBoNT/A. A prevalence of NABs of up to 13.9% after long-term aboBoNT/A or onaBoNT/A treatment of more than 10 years has been described [8,9].

Since 1995 rimabotulinumtoxinB (rimaBoNT/B) has also become available for the treatment of patients with CD [10,11]. Double-blind placebo-controlled studies have demonstrated safety and efficacy for both BoNT/A-responsive [12] and BoNT/A-resistant patients [13]. In a double-blind, randomized study [14], no difference in the responses to BoNT/A and BoNT/B was found in CD patients still responding well to BoNT/A. Pappert et al. [15] reported the results of a double-blind, randomized trial comparing BoNT/A and BoNT/B treatment in toxin-naïve patients. Similar to the study by Comella et al. [14] the latter study revealed the non-inferiority of the clinical effect after BoNT/B treatment in comparison to BoNT/A treatment.

In these studies, the effect of a single injection of rimaBoNT/B was analyzed. However, we were interested to compare the effect of repetitive injections of BoNT/A and BoNT/B not in different cohorts but in the same subjects. BoNT/A and BoNT/B are different botulinum toxin serotypes and cleave different proteins. BoNT/A cleaves SNAP-25 and BoNT/B cleaves VAMP [16,17,18]. It cannot be excluded that a patient responding well to a standard dose of BoNT/A only poorly responds to a standard dose of BoNT/B. Nothing seems to be known about the intracellular concentration of SNAP-25 and VAMP or their ratio in animal and human neurons and the variability of these concentrations across patients.

As a first approach to analyze the intraindividual comparison of BoNT/A and BoNT/B injections and the question of whether rimaBoNT/B injections may be an alternative to deep brain stimulation in CD patients with a STF after BoNT/A treatment we compared the response to the first four injections of BoNT/A and the response to the first four injections of BoNT/B in 17 patients with CD having become non-responsive to aboBoNT/A.

## 2. Results

### 2.1. Demographical and Treatment-Related Baseline Values

In the present cohort of CD patients the female/male ratio was low (10/7 = 1.4). The mean age at onset of BoNT/A therapy was 53 ± 7 years. Duration of BoNT/A therapy until STF became clinically manifest covered a wide age range of 1 to 12 years (Table 1).

Mean initial TSUI score at the onset of BoNT/A therapy was slightly higher than the mean TSUI score at the onset of BoNT/B therapy (Table 1). However, the variation of the TSUI scores at the time of switch to BoNT/B was much higher than at the onset of BoNT/A therapy. In 13 out of 17 patients the initial BoNT/B dose was 10 times the BoNT/A dose. Therefore, the mean initial dose of aboBoNT/A was close to the mean initial dose of rimaBoNT/B divided by 10 (Table 1).

### 2.2. Side Effects of the First Four BoNT/A and BoNT/B Injections

After the first BoNT/A injection, 6 out of 17 patients reported side effects: 3 patients reported neck weakness, 2 patients reported swallowing problems and 1 patient claimed on dry mouth. After the first BoNT/B injection 10 out of 17 patients reported side effects—8 patients had a mild to moderate dry mouth for about 4 weeks and 2 patients reported relevant constipation. With duration of BoNT/A therapy and in parallel to the reduction of total dose, the frequency of side effects declined. This was not the case after the second and third BoNT/B injection which had been performed with an increased dose. But after the fourth injection only 5 patients claimed to have a dry mouth and only 1 patient had a relevant constipation, although the mean BoNT/B dose had been further increased.

### 2.3. Comparison of the First BoNT/A and BoNT/B Injection in A Single Subject

In one (male; 53 y) of our 17 CD patients, the effect of the first injection of BoNT/B was monitored by his wife (PGA; Figure 1; open circles). She was trained to score her husband’s head position, his complaints of pain, and his handicap during everyday life activities by one single number as a percentage of the value she had determined before injection therapy was started. At the baseline visit (BV0), before BoNT/B therapy was started, she mentioned that she had done this scoring procedure already at the beginning of the BoNT/A treatment. The document was found in the old chart. She did not have a copy; thus, she was unable to compare her scoring after the first BoNT/B injection with the scores she had made years before after the first BoNT/A injection.

In Figure 1 her scores after the first injection with 1000 MU aboBoNT/A (full circles) and the first injection with rimaBoNT/B (7500 U NeuroBloc^®^/MyoBloc^®^; open circles) are presented. This patient responded slightly better to the fairly high dose of aboBoNT/A than to the fairly low dose of rimaBoNT/B. After the first BoNT/A injection, the improvement of symptoms started earlier, was slightly more pronounced, and lasted slightly longer than after the first BoNT/B injection (Figure 1). For both injections, the peak effect (33% after BoNT/A; 25% after BoNT/B) occurred between days 45 and 50.

The similarity of both curves is striking. The line in Figure 1 demonstrates that the clinical effect of the first BoNT/A and BoNT/B injection exceeded 84 days. When an injection is performed at that time the patients starts from a better situation than before. This implies that with repetitive injections every 12 weeks, a continuous improvement may occur and the TSUI score, when determined every 12 weeks just prior to the next injection may show a continuous decline (stair-case effect) as demonstrated in Figure 2A for the entire cohort.

### 2.4. Comparison of the BoNT/A and BoNT/B Injections in the Cohort

The patients’ subjective assessment of the clinical effect revealed a continuous improvement. Improvement of mean PGA increased from 25% after the first injection to 45% after the second BoNT/A injection. Improvement of mean PGA after the first BoNT/B injection was 22% and after the second injection was 42%. PGA of the next two BoNT/A injections showed a further improvement up to 52%, whereas for the next two BoNT/B injections a mild decrease of improvement down to 30% was observed (Table 2). Patients’ assessments were significantly different (*p* < 0.05) after four BoNT/A and four BoNT/B injections (Table 2).

The treating physician´s rating of the improvement by means of the TSUI score [19] revealed a similar time course (Figure 2A). The relative TSUI scores decreased significantly from the 100% baseline level, not only after the first two BoNT/A injections (*p* < 0.01; lower crosses in Figure 2A) but also after the first two BoNT/B injections (*p* < 0.05; upper crosses in Figure 2A). The difference in the level of significance results from the larger variability of the baseline scores at the time the patients were the first time injected with BoNT/B (see error bars in Figure 2A).

After the next two BoNT/A injections severity of CD further decreased (Figure 2A; full circles) although mean treatment cycle duration was significantly (*p* < 0.05) increased for further two weeks (Table 2), and the dose per session was significantly (*p* < 0.05) decreased from the initial mean dose of about 850 MU aboBoNT/A to about 750 MU (Figure 2B; full circles).

After the third and the fourth BoNT/B injection mean severity of CD started to worsen again (Figure 2A; open circles) although the length of the inter-injection interval was kept constant (Table 2) and mean dose of BoNT/B was significantly (*p* < 0.05) increased from 8500 U to more than 10,000 U (Figure 2B). The difference of the relative improvement between BoNT/A and BoNT/B injections became significant (*p* < 0.05; large star in Figure 2A) after four injections as the patient’s global assessment (Table 2).

After the fourth injection, three patients and their treating physician did not notice any relevant improvement; they had also developed a CSTF after BoNT/B therapy. In a further five patients, a PSTF had developed with a mild to moderate reduction of improvement.

### 2.5. Comparison of the Distribution of Relative Improvement after Two BoNT/A and Two BoNT/B Injections

Mean values of the relative improvement of CD determined by means of the TSUI score across all 17 patients after the first and the second BoNT/A and BoNT/B injections were nearly identical (Figure 2A). When the spectrum of relative improvement was subdivided into five different ranges (>−25 to 0%, >0 to 25%, >25 to 50%, >50 to 75%, and >75%) the distribution of relative improvement of CD after two injections of BoNT/A (Figure 3; full bars) and after two BoNT/B injections (open bars) could be compared and did not reveal a significant difference.

The distribution of the responses to BoNT/A non-significantly tended to be broader than to BoNT/B. Underlying individual values of visits AV2 and BV2 of Figure 3 are presented in Figure 4.

### 2.6. Correlation of the Relative Improvement of CD after Two BoNT/A and Two BoNT/B Injections

Despite of the same temporal development of the mean remaining severity of CD after two BoNT/A and BoNT/B injections (Figure 2A) and the similarity of the distributions of the relative improvement after 6 months of BoNT/A or BoNT/B treatment (Figure 3) no significant correlation was found when the response of a single patient to a BoNT/A injection was correlated with his response to the corresponding BoNT/B injection. This was the case for all four visits (V1, …, V4). As a typical example, the responses after the second BoNT/A injection determined at AV2 are plotted against the responses after the second BoNT/B injection at BV2 (Figure 4).

This result demonstrates that the response to BoNT-B injections cannot be predicted from the previous BoNT/A treatment.

## 3. Discussion

### 3.1. Demographic Data and Treatment-Related Data at Baseline Visits

In the present study a small cohort of CD patients is analyzed. Age at onset of symptoms was typical but compared to larger studies with a female/male ratio of 1.6 to 1.9, males were overrepresented. Initial aboBoNT/A dose was much higher than the starting dose of 500 U recommended in the SPC for aboBoNT/A (SPC, 08/04/2015). During the following treatments aboBoNT/A dose was significantly reduced. These initial high aboBoNT/A doses had been a relevant risk factor for the development of a STF later on [8,9,20]. Compared to another study on treatment with BoNT/B of patients with STF after BoNT/A treatment [13] and the recommended initial doses in the SPC (26/02/2014) the initial doses of rimaBoNT/B in the present study were comparatively low (Table 1).

### 3.2. Side Effects of the First Four BoNT/A and BoNT/B Injections

Frequency and intensity of side effects after BoNT/A and BoNT/B injections were within the range observed in other studies on BoNT/A or BoNT/B treatment [11,12,13,14,15]. A decline of the frequency of side effects with ongoing therapy was observed after BoNT/A injections in parallel to the decline in BoNT/A doses. Increase of BoNT/B dose is probably the reason why this decline did not occur during the first three BoNT/B injections. The clear decline in the frequency of side effects observed after four injections goes along with the decline of efficacy assessed by patients and the treating physician.

### 3.3. The Clinical Efficacy of BoNT/A and BoNT/B Injections

The present open-label study intraindividually comparing the clinical efficacy of BoNT/A and BoNT/B injections revealed that for the chosen doses, no significant difference was found after the first two to three injections (Figure 2A and Figure 3). This is not only the case for a single injection (Figure 1) but also for repetitive injections (Figure 2A and Figure 3). This finding is consistent with two larger double-blind, randomized trials demonstrating the non-inferiority of BoNT/B to BoNT/A for a single injection in toxin-naïve patients [15] and patients responding well to BoNT/A [12]. The doses used in the study by Pappert et al. [15] were 150 MU Botox^®^ versus 10,000 MU NeuroBloc/MyoBloc^®^. Assuming that 100 MU Botox^®^ are clinically equivalent to 300 Dysport^®^ [21] the initial BoNT/A doses in our study were much higher (corresponding to Botox^®^ doses of more than 200 MU) and the initial BoNT/B doses were lower than in the Pappert et al. study [15]. Nevertheless, no significant difference in the response to BoNT/A and BoNT/B was found in the present study after the first two injections.

A closer look at both studies shows that there is a non-significant tendency for better results in the BoNT/B arm in the Pappert et al. study [15], whereas in our study, a tendency for better results with BoNT/A injections was seen. A ratio of Dysport^®^ to Botox^®^ of 3.5 to 1 would imply that we had used a ratio of NeuroBloc/MyoBloc^®^ to Botox^®^ of 45:1 which is close to the ratios (40:1 and 50:1) used by two other previous studies comparing the effect of BoNT/A and BoNT/B [14,22] and lower than the ratio of 66.6:1 used in the Pappert et al. [15] study.

### 3.4. Difference in Efficacy of BoNT/A and BoNT/B Injections

Compared to the patients in previous studies, patients in the present study had a high complexity. They had aboBoNT/A-induced NABs and had developed a STF. Such patients do not respond as well to the second and third BoNT/A injection as CD patients who do not develop STF later on [23]. Comella et al. [14] treated patients still responding well to BoNT/A, Pappert et al. [15] analyzed toxin-naïve patients. It is interesting to see that the responses to BoNT/B and BoNT/A did not differ in the present cohort of complex patients. It may very well have been that the effect of BoNT/B injection had been even larger when our patients had been injected as toxin-naïve patients. Nevertheless, equal responses to BoNT/B and BoNT/A injections in patients with a secondary treatment failure and detectable titers of NABs against BoNT/A in the mouse hemidiaphragm assay (MHDA; [24,25]) is a clinical hint that no or little cross-immune response to BoNT/B occurs in patients having become resistant to BoNT/A.

Although the inter-injection intervals were kept constant during the BoNT/B treatment and the BoNT/B doses per session were continuously increased, the improvement of CD declined again after the third injection in contrast to the improvement after three to four BoNT/A injections for which the inter-injection intervals were significantly increased (Table 2) and the doses per session decreased (Figure 2B). The significant difference in the outcome between BoNT/A and BoNT/B treatment after four injections (Figure 2A) indicates that in the CD patients with an STF after aboBoNT/A therapy, the beginning of partial secondary treatment failure after rimaBoNT/B therapy had to be also suspected. After two BoNT/B injections, all patients showed at least as good a response as after two BoNT/A injections (Figure 2A and Figure 3). However, after the third injection, one patient (1/17 = 5.9%) reported that he did not have any response. Further, two patients (2/17 = 11.8 %) reported a moderate reduction of efficacy of about 50%. After the fourth injection three patients (17.6%) developed a CSTF and five patients a PSTF (29.4%). In more than half of the patients, however, a satisfactory response persisted.

These findings are consistent with the observation by Dressler et al. [26], of an improvement of about 40% after the first two rimaBoNT/B injections in 10 CD patients with high titers of NABs against BoNT/A. But in their cohort 4 out of 10 patients (40%) developed CSTF after the third and 6 out of 9 patients (67%) after the fourth injection [26]. This difference in the frequency of the development of CSTF and PSTF (and in the frequency of side effects) can be explained by the difference in dose of rimaBoNT/B per session. In the present study mean dose per session was increased from 8480 ± 2510 to 10,200 ± 2300, whereas in the study by Dressler et al. [26], BoNT/B was increased from 12,370 ± 1804 to 12,972 ± 2868.

### 3.5. Lack of Correlation between the Response to BoNT/A and BoNT/B

The finding of a lack of an intraindividual correlation between the responses to BoNT/A and BoNT/B is based on a rather small patient group. Nevertheless, it was consistently found for all four visits, V1 to V4. Since BoNT/A cleaves SNAP-25 and BoNT/B cleaves VAMP [16,17,18], the effect of both toxins is difficult to compare. We know that the binding efficacy of BoNT/B to human neurons is low [27], but in clinical practice such differences in binding efficacy can be compensated by the use of higher doses. Another question is whether there is a fixed ratio of SNAP25 and VAMP in human neurons or whether this ratio varies due to genetic reasons. Even in case of a fixed ratio of SNAP25 and VAMP concentration in human neurons, no fixed ratio of cleavage of SNAP-25 and VAMP and no correlation between the responses to BoNT/A and BoNT/B can be expected after the use of standard doses of BoNT/A and BoNT/B. Also, the uptake of BoNT/A and BoNT/B may independently vary from patient to patient. Because of such unsolved problems it is highly likely that no correlation of responses to standard BoNT/A and BoNT/B doses exists, as demonstrated in the present study.

### 3.6. Implications for Patients with a Secondary Treatment Failure

The clinical implication of the lack of correlation between the responses to BoNT/A and BoNT/B is that the response to BoNT/B cannot be predicted from the response to BoNT/A. For patients with a complete STF to BoNT/A switching to BoNT/B is not a long-term alternative because of the high risk of developing another STF against BoNT/B. Even the short-term response to BoNT/B cannot be predicted from the previous BoNT/A therapy in an individual case. For a patient with CSTF waiting to be operated by deep brain stimulation, a switch to BoNT/B may be considered to bridge the time to operation.

It has been recommended that patients with STF undergo deep brain stimulation (DBS) or terminate BoNT/A therapy because NAB titers decline after cessation of therapy [28]. However, it takes time until patients with a NAB-induced STF become NAB negative again—30 months was the mean (range: 10 to 72 months) when the MPA was used to control the immune status [29], and 1895 ± 1211 days (corresponding to 63 ± 40.4 months) when the more sensitive MHDA was used [30,31]. Restarting with the same BoNT/A preparation bears the risk of an reactivation of the immune response [29,31]. This does not seem to be the case when incoBoNT/A is used for the restart [31]. In our experience it is not necessary to wait until a patient becomes NAB-negative again before he is switched to incoBoNT/A. NAB titers may also decline when patients are switched to incoBoNT/A immediately after a positive NAB test [32]. However, decline of NAB titers under ongoing incoBoNT/A appears to be variable and may last more than 4 years until half of the MHDA-positive patients have become negative again [32]. Furthermore, the long-term clinical outcome after restarting with incoBoNT/A is so far unclear. It will be interesting to see whether switching to incoBoNT/A is also an alternative, not only for those patients with STF after the usual STF risk factors such as high dose-per-session and long duration of treatment (for a recent review see [33]), but also for the special subgroup of patients with a genetic predisposition to BoNT-antibody formation [34]. And it will be interesting to see whether there is a special subgroup of patients with a satisfactory long-term outcome under rimaBoNT/B therapy without NAB induction.

### 3.7. Strengths and Limitations of the Study

The strength of this purely observational study is that for the first time the responses to BoNT/A and BoNT/B are compared on an individual basis. The lack of correlation between the responses to BoNT/A and BoNT/B was a consistent finding in this small open-label study. A limitation of the study was that the switch from BoNT/A to BoNT/B had not been performed according to a fixed protocol. It will be interesting to see whether the results of the present study can be replicated in a double-blind cross-over long-term study with the opportunity to compare the temporal development of NABs against BoNT/A and BoNT/B.

## 4. Conclusions

In this study it was demonstrated that CD patients with an STF and NABs against BoNT/A respond to rimaBoNT/B injections. However, rimaBoNT/B injections offer only a short-term improvement in the mean, but not a long-term perspective because of the high risk for the development of a secondary immune response to rimaBoNT/B. Unfortunately, the response to BoNT/B cannot be predicted from the response to BoNT/A.

## 5. Materials and Methods

### 5.1. Patients

This observational study was performed according to the guidelines of good clinical practice (GCP) and the Declaration of Helsinki.

Inclusion criteria were: (i) age > 18, (ii) diagnosis of idiopathic cervical dystonia, (iii) onset of BoNT/A therapy in our center, (iv) complete documentation (date of injection, preparation used, total dose, muscles injected, dose per muscle, TSUI score at the day of injection, patient´s global assessment of the severity of CD (PGA) at the day of an injection) of the first five BoNT/A injections, (iv) documented development of a secondary treatment failure (STF) according to the criteria of our center (for details see below [23]) and a confirming positive MHDA test [25], and (v) complete documentation of the first five BoNT/B injections. 

Excluded were patients with an injection cycle longer than 4 months during the first year of BoNT/A or BoNT/B treatment.

Out of 543 charts screened, only 17 charts met all inclusion and exclusion criteria.

### 5.2. BoNT Injections

From the charts’ data at the onset of BoNT/A therapy (visit 0; AV0) and the next four BoNT/A injections (at visits 1 to 4; AV1, AV2, AV3, and AV4) were extracted. Baseline visit (BV0) for the BoNT/B therapy was the day of switch to BoNT/B. Data of this visit (BV0) and the next four visits (BV1, …, BV4) were extracted from the charts.

All 17 patients had only received abobotulinumtoxinA (aboBoNT/A; Dysport^®^)(SPC 08/04/2015) injections before they were switched to rimabotulinumtoxinB (rimaBoNT/B; NeuroBloc/MyoBloc^®^)(SPC 26/02/2014). For aboBoNT/A injections 500 U were diluted in 2.5 mL, for rimaBoNT/B injections 5000 U were also diluted in 2.5 mL. In 13 out of 17 patients the last injection scheme for BoNT/A injections was also used for BoNT/B injections. RimaBoNT/B dose per muscle was determined by multiplying the aboBoNT/A dose by 10. As for the BoNT/A injections the treating physician had been free to modify the injection scheme, dose per muscle, and the duration of the treatment cycle of the BoNT/B injections.

### 5.3. Criteria for the Development of STF and Antibody Testing

STF was suspected when patients had reported a highly-reduced effect and the treating physician had determined an increase of the TSUI score of more than 3 points during the last two injection cycles (for further details see [23]).

Blood samples had been taken, coded, and sent to an independent, blinded contractor (Toxogen^®^, Hannover, Germany) to be tested for the presence of neutralizing antibodies by means of the MHDA. Months after the switch to BoNT/B the Toxogen^®^ laboratory returned a list with the test results of all 17 samples.

### 5.4. Assessment of BoNT Injections by Patients and Treating Physician

Patients or relatives of patients assessed the remaining severity of CD after a BoNT/A injection using a global assessment scale (PGA) ranging from 0 to 100, where 100 corresponded to the severity of CD at onset of BoNT/A therapy. An example is presented in Figure 1. The response to BoNT/B was assessed similarly with 100 corresponding to the severity of CD at BV0. The difference (100-PGA) was the improvement assessed by the patients.

The severity of CD was also scored by the treating physician using the TSUI score [19]. In our department, the TSUI score is determined immediately before each BoNT injection. For sake of comparison, the remaining severity of CD was determined as a percentage of the mean baseline score. All TSUI scores before and after BoNT/A injections were divided by the mean baseline TSUI score at AV0, all TSUI scores before and after BoNT/B injections were divided by the mean value of the TSUI score at day BV0 of the switch to BoNT/B. For all patients, relative improvement was calculated as the difference between each score and the corresponding baseline score as a percent of this baseline score).

The spectrum of relative improvement was subdivided into five different ranges (>−25 to 0%, >0 to 25%, >25 to 50%, >50 to 75%, and >75%). The distributions of the relative improvement after BoNT/A and after BoNT/B therapy were determined for all 4 visits (V1, …, V4).

### 5.5. Statistics

The relative improvements under BoNT/A and BoNT/B treatment at V1 to V4 were non-parametrically correlated (rank correlation; Figure 4). Relative improvement and unified doses were compared non-parametrically (U-test with alpha-adjustment; Figure 2A,B). Distributions after BoNT/A injections were compared with distributions after BoNT/B injections by means of the Friedman test (Figure 3). Statistical tests were part of the SSPS statistics package (version 25; IBM; Armonk, NY, USA).

### 5.6. Statement of Ethics

This study was approved by the local ethics committee of the Heinrich-Heine University Duesseldorf, Germany (5 April 2013, study number: 4085). In accordance with the Declaration of Helsinki, written informed consent was obtained from all patients at the outpatient clinic.

## Figures and Tables

**Figure 1 toxins-12-00677-f001:**
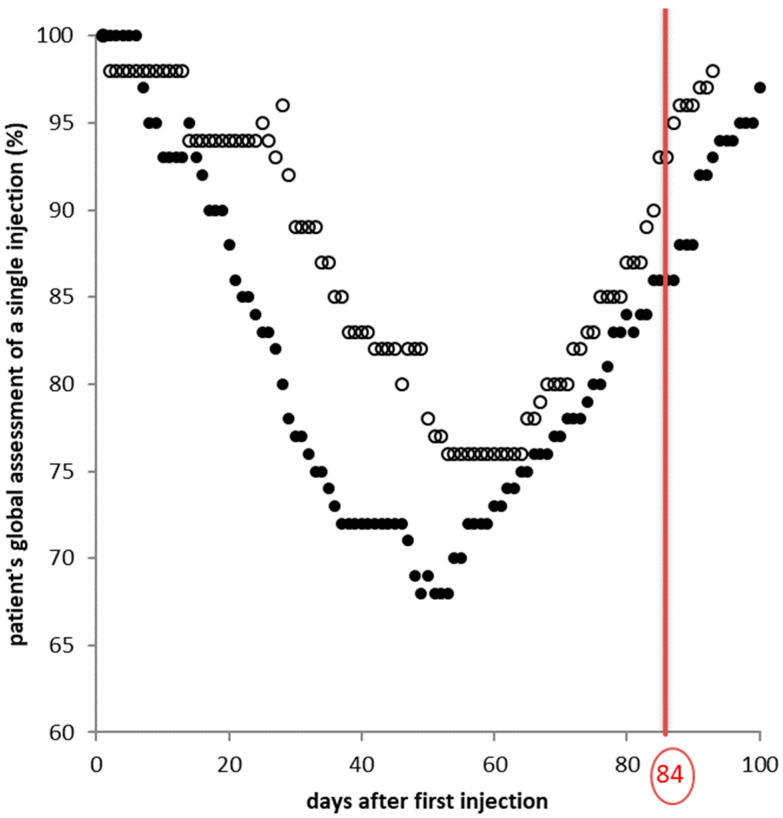
Daily PGA scores of cervical dystonia (CD) in a single patient. Full circles—after the first BoNT-A injection; open circles—after the first BoNT-B injection. PGA is patient´s global assessment of the severity of CD as a percent of the severity of CD at baseline visit V0. The vertical line indicates day 84 (= 12 weeks) after V0.

**Figure 2 toxins-12-00677-f002:**
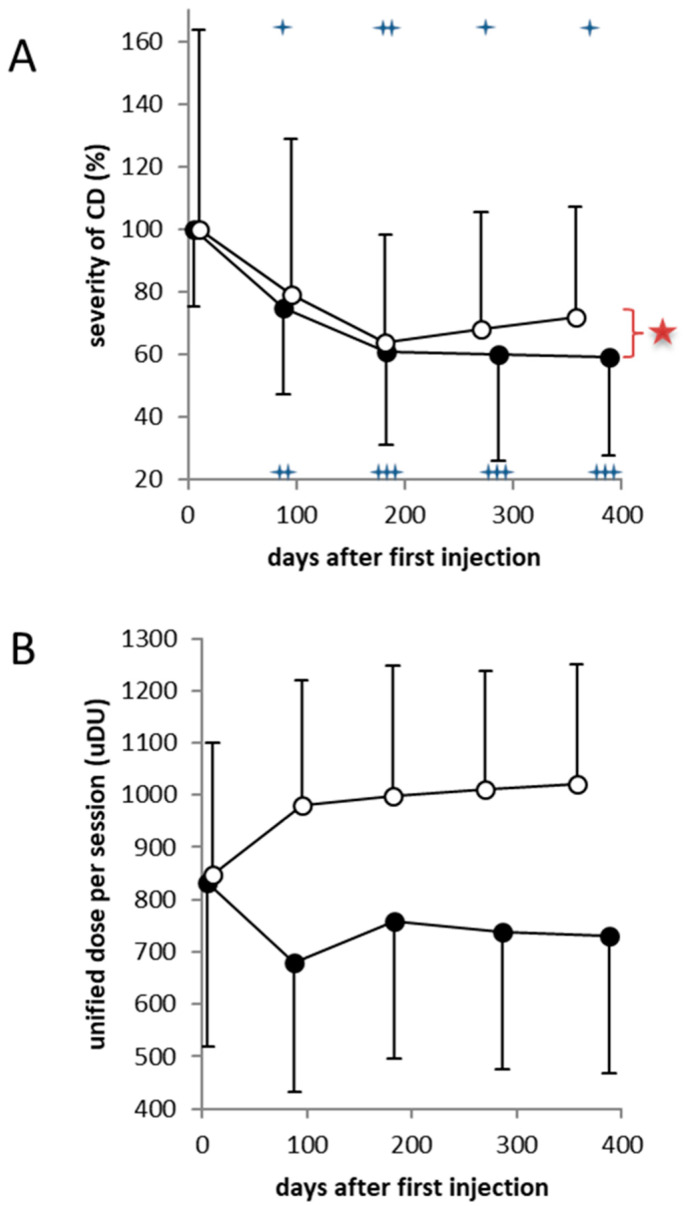
(**A**) Significant continuous decline of the remaining severity of CD as a percentage of the baseline severity after the first four abobotulinumtoxinA (aboBoNT/A) injections (full circles) and after the first four BoNT/B injections (open circles). The variability of the severity of CD was larger when the patients were injected with BoNT/B (upper error bars) than when they were injected with botulinum toxin type A (BoNT/A) (lower error bars). The inter-injection intervals could be significantly increased (Table 2). Severity of CD significantly decreased (+ = *p* < 0.05; ++ = *p* < 0.01, +++ = *p* < 0.001). During BoNT/B therapy severity of CD significantly decreased (upper crosses: + = p < 0.05; ++ = *p* < 0.01). The duration of inter-injection intervals were kept constant. (**B**) When BoNT/B therapy was started, the unified dose per session was adjusted to the initial unified dose of BoNT/A therapy. With duration of therapy, BoNT/A doses were significantly decreased and BoNT/B doses significantly increased. Unified dose units were determined by dividing the rimaBoNT/B dose by 10.

**Figure 3 toxins-12-00677-f003:**
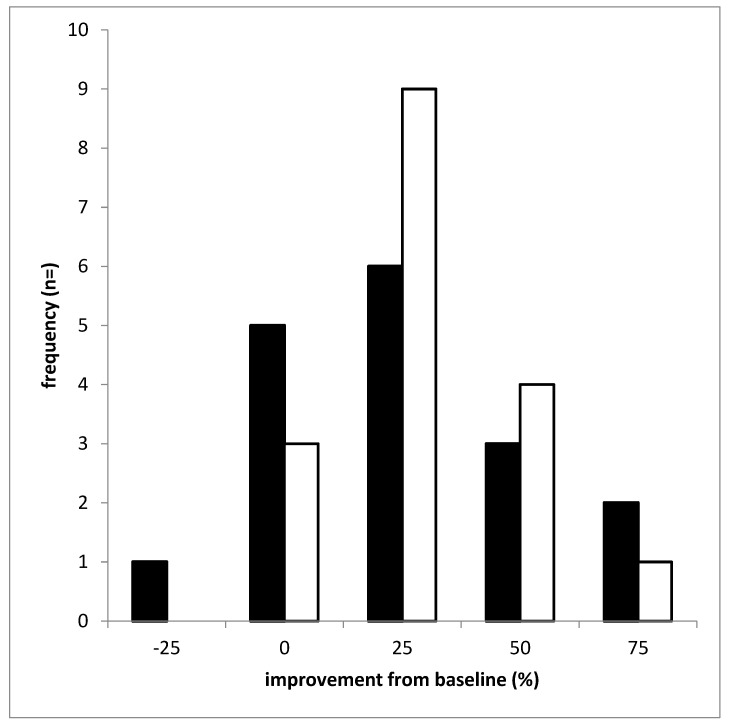
Distribution of the relative improvements based on the TSUI scores after two injections of BoNT/A (full bars) in comparison to the relative improvements after two injections with BoNT/B (open bars). The spectrum of relative improvements is subdivided into five different percentage ranges (x-axis; for details see Methods). On the y-axis the number of patients per improvement range is presented.

**Figure 4 toxins-12-00677-f004:**
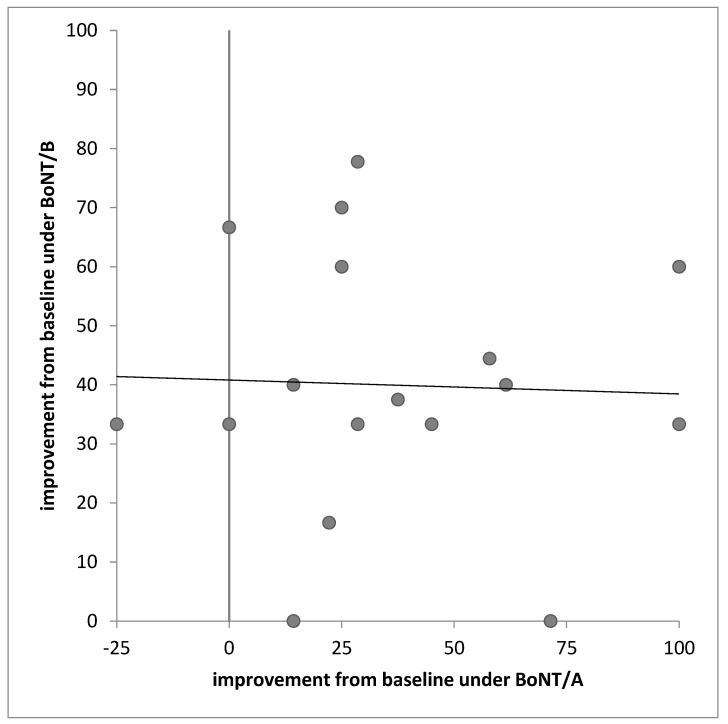
No correlation was found between the relative improvements based on the TSUI scores after two BoNT/A injections at visit AV2 (*x*-axis) and the relative improvements after two BoNT/B injections at visit BV2 (*y*-axis).

**Table 1 toxins-12-00677-t001:** Demographic and baseline treatment-related data:

Parameters	Mean/S.D.
Gender distribution	10 females, 7 males
Age at onset of BoNT/A therapy	53.0/7.0 years
Age at onset of BoNT/B therapy	58.7/6.8 years
Duration of BoNT/A therapy	5.7/3.5 years
TSUI at onset of BoNT/A therapy	9.5/2.1
TSUI at onset of BoNT/B therapy	8.8/4.2
Initial dose of aboBoNT/A	832/314 U
Initial dose of rimaBoNT/B	8480/2510 U

**Table 2 toxins-12-00677-t002:** Mean patients’ assessment of the efficacy of injection 1 to 4 and durations of cycles 1 to 4.

Parameter	Mean/S.D.	Significance (*p*-Level)
Improvement of PGA at AV1 Improvement of PGA at BV1	25/20 22/23	n.s.
Improvement of PGA at AV2 improvement of PGA at BV2	45/15 42/17	n.s.
Improvement of PGA at AV3 Improvement of PGA at BV3	50/25 40/22	n.s.
Improvement of PGA at AV4 Improvement of PGA at BV4	52/20 30/19	*p* < 0.05
Duration of cycle 1 of BoNT/A therapy Duration of cycle 1 of BoNT/A therapy	89/12 92/12	n.s.
Duration of cycle 2 of BoNT/A therapy Duration of cycle 2 of BoNT/A therapy	93/13 91/12	n.s.
Duration of cycle 3 of BoNT/A therapy Duration of cycle 3 of BoNT/A therapy	100/15 89/14	n.s.
Duration of cycle 4 of BoNT/A therapy Duration of cycle 4 of BoNT/A therapy	107/17 88/12	*p* < 0.05

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
