# Peer review of "Transient Improvement after Switch to Low Doses of RimabotulinumtoxinB in Patients Resistant to AbobotulinumtoxinA"

_toxins, 2020, doi:10.3390/toxins12110677_

Round 1

Reviewer 1 Report

Manuscript entitled “Transient improvement after switch to low doses of rimabotulinumtoxin B in patients resistant to abobotulinumtoxin A” refers on the possible use of BoNT/B as an alternative for patients affected by cervical dystonia having become resistant to BoNT/A.

The topic is interesting, and data reported in the manuscript could be of great interest for physicians that use botulinum toxins for therapeutic purposes.

The introduction is well written and correctly reports the state of the art.

I have some concerns respect to the other paragraphs:

-          Materials and methods:

Materials and methods should allow to other researchers the replication of experiments and so this section should describe criteria adopted for patients’ enrollment, BoNT injections procedures, criteria adopted for the assessment of BoNT effectiveness, statistical analysis.

Authors should consider the opportunity to reword paragraph 2.1 as Patients; “and BoNT/A injections” should be deleted. If data reported in fig. 2B are relevant for materials and methods, perhaps they should be shown as table, however if Authors prefer the figure, this should be fig. 1 (because it is the first figure that appear in the text).

In paragraph 2.3 is reported the assessment of BoNT injections. Patients or their relatives assessed the remaining severity of cervical dystonia after BoNT injection. Authors should detail the criteria adopted to assess this severity. 100 corresponded to the severity of cervical dystonia on the day of the first injection. What are the criteria to assign other values? In fig 1 Authors presented an example of the assessment performed by the wife of a patient. If this data is relevant Authors should include also the assessment performed by other patients or their relatives, otherwise this information should be deleted.

-          Results:

If enrolled patients are those that received BoNT/A and become resistant to this treatment, data on BoNT/A injection are not significant as result but are a pre-requisite. Authors should consider the opportunity of moving this information in a table under materials and methods paragraph or comment it in the discussion paragraph.

Is paragraph 3.2 relevant? If yes, why are reported only data on a patient? Please consider the opportunity to delete or implement this paragraph.

-          Discussion:

In this section are reported comments on obtained results. With the aim of make the manuscript easier for reader Authors should consider they opportunity to use the same schema adopted in results section.

-          Conclusion:

I think that limitations of the study should be presented in the first part of this section. Authors should consider the opportunity to include in conclusion also some sentences on the strengths of their work.

If relevant, Figure 4 should be presented in results and commented in conclusion.

Are the conclusions reported in the text the same of those reported in the abstract?

Reading the manuscript, I am afraid that Authors erroneously submitted a draft and not the final version of their manuscript. 

In my opinion this manuscript cannot be published in Toxins without an overall revision.

Author Response

Reviewer: 1, 2 and 3

General remark to all three reviewers:

Unfortunately, the final version was not submitted. This can be seen from the fact, that the incomplete Materials and Methods section followed the introduction and not as prepared for Toxins after Discussion and Conclusions. That caused a lot of trouble: sequence of Figures etc.

We apologize for that. We are very thankful to the reviewers that they did not turn down the manuscript because of these shortcomings.

Manuscript entitled “Transient improvement after switch to low doses of rimabotulinumtoxin B in patients resistant to abobotulinumtoxin A” refers on the possible use of BoNT/B as an alternative for patients affected by cervical dystonia having become resistant to BoNT/A.

The topic is interesting, and data reported in the manuscript could be of great interest for physicians that use botulinum toxins for therapeutic purposes.

All 3 reviewers have substantially contributed to the improvement oft he revised manuscript.

The authors are thankful for that. 

The introduction is well written and correctly reports the state of the art.

I have some concerns respect to the other paragraphs:

Details have been corrected in the introduction as suggested by reviewer 2, but in general the introduction remained as before.

-          Materials and methods:

Materials and methods should allow to other researchers the replication of experiments and so this section should describe criteria adopted for patients’ enrollment, BoNT injections procedures, criteria adopted for the assessment of BoNT effectiveness, statistical analysis.

The submitted manuscript did not contain a detailed section „Materials and Methods“.

In the revised version of the manuscript „Materials and methods“ is placed where it should be and much more details are presented.

Authors should consider the opportunity to reword paragraph 2.1 as Patients; “and BoNT/A injections” should be deleted. If data reported in fig. 2B are relevant for materials and methods, perhaps they should be shown as table, however if Authors prefer the figure, this should be fig. 1 (because it is the first figure that appear in the text).

The figures are now presented in the correct order.

We have added two tables. More details on doses are now presented in Tab. 1.

In paragraph 2.3 is reported the assessment of BoNT injections. Patients or their relatives assessed the remaining severity of cervical dystonia after BoNT injection. Authors should detail the criteria adopted to assess this severity. 100 corresponded to the severity of cervical dystonia on the day of the first injection. What are the criteria to assign other values?

In fig 1 Authors presented an example of the assessment performed by the wife of a patient.

If this data is relevant Authors should include also the assessment performed by other patients or their relatives, otherwise this information should be deleted.

It was explained to the patient that he had to use a scale from 0 to 100 to score the remaining severity of his disease in %  of the severity at onset of BoNT/A therapy. As for a visual analogue scale no further instruction were made.

Each patient scored severity of CD at visits AV0, AV1, AV2, AV3, and AV4. and at BV0, BV1, BV2, BV3 and BV4.

It was explained to the patient that severity of CD at visit 0 was 100%.

Mean PGA is presented in Tab. 2 for visits 1, 2, 3 and 4.  

-          Results:

If enrolled patients are those that received BoNT/A and become resistant to this treatment, data on BoNT/A injection are not significant as result but are a pre-requisite. Authors should consider the opportunity of moving this information in a table under materials and methods paragraph or comment it in the discussion paragraph.

Tab.1 contains information on initial severity of CD at onset of BoNT/A and BoNT/B therapy as well as initial dose of BoNT/A and BoNT/B.

Is paragraph 3.2 relevant? If yes, why are reported only data on a patient? Please consider the opportunity to delete or implement this paragraph.

These data together with Tab. 2 demonstrate that

1.       patient´s scoring is highly reliable

2.       duration of BoNT/A and BoNT/B may exceed 12 weeks

3.       the analysis of PGA gives useful information on secondary treatment failure

-          Discussion:

In this section are reported comments on obtained results. With the aim of make the manuscript easier for reader Authors should consider they opportunity to use the same schema adopted in results section.

In the revised manuscript the same flow of thoughts is now presented as in the result section.

Authors are thankful for this remark.

-          Conclusion:

I think that limitations of the study should be presented in the first part of this section. Authors should consider the opportunity to include in conclusion also some sentences on the strengths of their work.

If relevant, Figure 4 should be presented in results and commented in conclusion.

Are the conclusions reported in the text the same of those reported in the abstract?

Limitations and strengths are both mentioned now.

Reviewer 1 is right.

Now abstract and conclusions are matched.

Reading the manuscript, I am afraid that Authors erroneously submitted a draft and not the final version of their manuscript. In my opinion this manuscript cannot be published in Toxins without an overall revision.

Reviewer 1 is absolutely right:

 Unfortunately the final version was not submitted (see general remarks).

We again apologize for that.

Reviewer 2 Report

This manuscript gives useful information for clinicians using BoNT.  However, considerable work is needed to revise the manuscript before it can be accepted for publication.

Introduction

Line 32                    Citations 2 and 3 are too old, over 30 years each.  More recent citations should be used to demonstrate that BoNT is a treatment of choice nowadays.

Line 36                    The abbreviations used include the word “Failure” (F) and not non-response.  This should be corrected.

Line 278                  Citation 5 is incomplete

Line 34                    Citation 5 reports an antibody prevalence rate of 4.9% to 7.1% in the population studied, as measured by a mouse neutralisation assay.  Citation 6 reports an antibody prevalence of 3% with an alternative test method.  Neither report the 40% cited by the authors.  This should be corrected.

Line 38                     I could find nothing in citation 7 which describes a change to the ONA manufacturing process.  A correct citation should be provided.  Also, the correct primary citation for 5ng of clostridial protein per 100 vial of ONA should be the US prescribing information:

https://media.allergan.com/actavis/actavis/media/allergan-pdf-documents/product-prescribing/20190620-BOTOX-100-and-200-Units-v3-0USPI1145-v2-0MG1145.pdf Section 11

Line 39                     Citation 8 gives and antibody prevalence rate for ONA in CD of 1.2% using a mouse neutralisation assay.  The authors should cite this rate.

Line 41                     Citation 9 gives an antibody prevalence rate of 18.4% as measured by ELISA and 14.6% as measured by a mouse hemdiaphragm assay for CD patients.  Citation 10 examined 5 different conditions and cites rates of antibody formation from 0% to 17.31%.  Neither of these citations quotes 13%, as given in the manuscript.  The authors should clear this up and cite the correct values for specified clinical conditions with specified assays (since the determination of neutralising antibodies is highly dependant on the assay method used).

Lines 41-43              It is incorrect to make this statement about ABO as this product does not have a comparable protein content to ONA on a per unit basis because the potency units are not the same.

Line 44                    Should be 1995, not 1997

Lines 46-48             This sentence needs to be rewritten

Line 55                    “attack” should be “cleaves”

Lines 55-57             How did the authors arrive at this statement?  Why would a different target for BoNT/B make a difference in clinical response?  The authors should clarify this.

Lines 57-59             The authors have already cited sufficient prior clinical studies to demonstrate that BoNT/A and BoNT/B give equivalent clinical responses in patients.  Therefore, this statement is incorrect in their context.  Also, they should cite publications which show their proposed genetic reasons for differences in the toxin targets.  Unless this has been shown, together with clear clinical evidence of differences in response to the 2 toxins in such patients, this statement should be removed.

Materials and Methods

Parts of the Materials and Methods section presents results.  The authors should rework this section to remove results and place them in the Results section.

Line 68                    The correct term for the abbreviation should be used – mouse hemidiaphragm assay (MHDA)

Line 68                    Citation 20 should be omitted as it is not applicable – different methodology and animal species used

Line 70                    The authors cannot use Figure 2 before Figure 1 or Figure 2b before Figure 2a.  They should correct their Figure order

Figure 2b                 The authors need to explain, in detail, the meaning of the “unified dose per session” and how this is used in their work.  The explanation needs to be very clear and thorough.

Legend to Figure 2a           This describes 4 injections but there are 5 shown, 4 after the initial injection.. Also, how can the reader see the data that supports the statement “The variability of the severity of CD was larger when the patients were injected with BoNT/B (upper error bars) than at the time they were injected with BoNT/A (lower error bars)” when these error bars look the same……

Lines 85-86             Citation 23 does not provide a recommendation of 1:10 ABO to RIMA “in full agreement with European consensus paper [23]”

According to the respective summary of product characteristics (SPC—last accessed 08/04/2015), the suggested starting total dose is 500 IU in two-three muscles, for abobotulinumtoxinA (SPC last text revision 11/12/2013),

For rimabotulinumtoxinB, an initial dose of 5,000 IU may be considered, but a dose of 10,000 IU divided between two and four muscles may be more effective (SPC, 26/02/2014).

The authors should moderate and clarify their statement.  Also, they should reference the SPCs of the products as primary citations and not citation 23.

Lines 91-93   This should be Figure 2 and not Figure 1 (see above comments)

Lines 102-103         Why have the authors suddenly presented an effect comparison between 2 injections when their work was focused on effect comparisons for five injections of each toxin?

Results

Section 3.1              A table of patient demographics should be included for clarity.

Lines 121-122                    Conversely, does this mean that the side effects for BoNT/A treated patients declined over time because the dose was decreased?  If so, the authors should state that.

Lines 137-142                    The authors discuss the results for the single patient in weeks, but the data presented in the corresponding figure are in days.  This difference should be corrected.

Line 148 et seq                  The authors discuss significance levels here.  These should also be shown directly on the data in the figure.

Lines 173-176                    Are the authors referring to Figure 4 here?

Discussion

Line 181                  Cannot refer to Figure 1 (single patient data) here together with the other figures as these data in Figure 1 are for one injection of either toxin alone.

Lines 185-186                    The ONA: ABO ratio of 1:3 or 1:4 stated here is not as recommended in citation 23.  See below.

Lines 190-191                    This statement about the Pappert study cannot be made as that was a non-inferiority study: superiority cannot be determined from such a study.

Line 197 (also previously)            The authors should describe and illustrate the staircase effect they refer to

Line 201                  The term “highly negatively selected” is strange and should be revised

Lines 206-206                    The present authors have not presented appropriate data on non-inferiority (other than observational) and so this statement cannot be made.

Lines 220-221         A reduction in efficacy of about 50% cannot be described as “moderate”

Lines 235-237                    The authors have made a statement about the ratio of toxin target cleavage but presented no data on this molecular aspect.  This is incorrect and based on an assumption about the clinical doses they have used.  What are the ratios of SNAP25 and VAMP in human neurons?  Is the efficacy of toxin uptake the same for types A and B?  We know already that the binding efficacy of type B to human neurons is very low (STROTMEIER, J., WILLJES, G., BINZ, T. & RUMMEL, A. 2012. Human synaptotagmin-II is not a high affinity receptor for botulinum neurotoxin B and G: increased therapeutic dosage and immunogenicity. FEBS Lett, 586, 310-3). This must be taken into account in any comments about target cleavage. 

Lines 238-241         Need to be supported by citations on the various issues the authors raise or this part should be removed.

Author Response

Reviewer 2

This manuscript gives useful information for clinicians using BoNT.  However, considerable work is needed to revise the manuscript before it can be accepted for publication.

The manuscript has substantially been revised according to reviewers´ comments.

Introduction

Line 32                    Citations 2 and 3 are too old, over 30 years each.  More recent citations should be used to demonstrate that BoNT is a treatment of choice nowadays.

Line 36                    The abbreviations used include the word “Failure” (F) and not non-response.  This should be corrected.

Line 278                  Citation 5 is incomplete

Line 34                    Citation 5 reports an antibody prevalence rate of 4.9% to 7.1% in the population studied, as measured by a mouse neutralisation assay.  Citation 6 reports an antibody prevalence of 3% with an alternative test method.  Neither report the 40% cited by the authors.  This should be corrected.

Line 38                     I could find nothing in citation 7 which describes a change to the ONA manufacturing process.  A correct citation should be provided.  Also, the correct primary citation for 5ng of clostridial protein per 100 vial of ONA should be the US prescribing information:

https://media.allergan.com/actavis/actavis/media/allergan-pdf-documents/product-prescribing/20190620-BOTOX-100-and-200-Units-v3-0USPI1145-v2-0MG1145.pdf Section 11

Line 39                     Citation 8 gives and antibody prevalence rate for ONA in CD of 1.2% using a mouse neutralisation assay.  The authors should cite this rate.

Line 41                     Citation 9 gives an antibody prevalence rate of 18.4% as measured by ELISA and 14.6% as measured by a mouse hemdiaphragm assay for CD patients. 

Citation 10 examined 5 different conditions and cites rates of antibody formation from 0% to 17.31%.  Neither of these citations quotes 13%, as given in the manuscript. 

The authors should clear this up and cite the correct values for specified clinical conditions with specified assays (since the determination of neutralising antibodies is highly dependant on the assay method used).

Lines 41-43              It is incorrect to make this statement about ABO as this product does not have a comparable protein content to ONA on a per unit basis because the potency units are not the same.

Line 44                    Should be 1995, not 1997

Lines 46-48             This sentence needs to be rewritten

Line 55                    “attack” should be “cleaves”

Lines 55-57             How did the authors arrive at this statement?  Why would a different target for BoNT/B make a difference in clinical response?  The authors should clarify this.

Lines 57-59             The authors have already cited sufficient prior clinical studies to demonstrate that BoNT/A and BoNT/B give equivalent clinical responses in patients.  Therefore, this statement is incorrect in their context.  Also, they should cite publications which show their proposed genetic reasons for differences in the toxin targets.  Unless this has been shown, together with clear clinical evidence of differences in response to the 2 toxins in such patients, this statement should be removed.

Only recent references are cited now.

Ok, correction is performed

Citation 5 is replaced by another article of the same group.

Kranz et al. 2008 reports >40% of PSTF and low titers of NABs. This reference is included now.

Brin et al. 2008 reports up to 17% NAB positive patients under the old BOTOX.

Brin et al. reports on a lower protein content of the new BOTOX. How can this be managed without change of  the manufacturing process?

We follow reviewer 2 and cite the US prescribing information 

We mention the 1.2% in the Brin et al. study.

HH is senior author of the citation 10 paper. We mentioned 13.9% there. 13% is corrected to 13.9% now in the revised manuscript.

Reviewer 2 is absolutely right:

The determination of neutralising antibodies is highly dependant on the assay method used. But comparison of antibody prevalence under different conditions is not topic of the present paper. Therefore we have not specified the different methods used for NAB measurements. We can easily provide the reader with more information on the different assays, but we will do this only if reviewer 2 thinks that this information is essential for the understanding of the present paper.

Reviewer 2 is right:

The difference in potency is well-known. We now mention the neurotoxin content of abo- and onaBoNT/A and the difference in potency per unit.

Is corrected.

Sentence has been rewritten.

Is corrected

Of course clinical responses can be made equal by adjustment of doses.

However, the question remains how many clostridial protein is needed for an equal clinical response.

The paragraph from line 55 to 59

Is completely rewritten.   

Materials and Methods

Parts of the Materials and Methods section presents results.  The authors should rework this section to remove results and place them in the Results section.

Line 68                    The correct term for the abbreviation should be used – mouse hemidiaphragm assay (MHDA)

Line 68                    Citation 20 should be omitted as it is not applicable – different methodology and animal species used

Line 70                    The authors cannot use Figure 2 before Figure 1 or Figure 2b before Figure 2a.  They should correct their Figure order

Figure 2b                 The authors need to explain, in detail, the meaning of the “unified dose per session” and how this is used in their work.  The explanation needs to be very clear and thorough.

Legend to Figure 2a           This describes 4 injections but there are 5 shown, 4 after the initial injection.. Also, how can the reader see the data that supports the statement “The variability of the severity of CD was larger when the patients were injected with BoNT/B (upper error bars) than at the time they were injected with BoNT/A (lower error bars)” when these error bars look the same……

Lines 85-86             Citation 23 does not provide a recommendation of 1:10 ABO to RIMA “in full agreement with European consensus paper [23]”

According to the respective summary of product characteristics (SPC—last accessed 08/04/2015), the suggested starting total dose is 500 IU in two-three muscles, for abobotulinumtoxinA (SPC last text revision 11/12/2013),

For rimabotulinumtoxinB, an initial dose of 5,000 IU may be considered, but a dose of 10,000 IU divided between two and four muscles may be more effective (SPC, 26/02/2014).

The authors should moderate and clarify their statement.  Also, they should reference the SPCs of the products as primary citations and not citation 23.

Lines 91-93   This should be Figure 2 and not Figure 1 (see above comments)

Lines 102-103         Why have the authors suddenly presented an effect comparison between 2 injections when their work was focused on effect comparisons for five injections of each toxin?

Reviewer 2 is right:

The section „Materials and Methods“ is now placed where they should be in a Toxins-paper. We have substantially rewritten this section.

This is corrected now.

We have omitted this citation.

We have corrected the order oft he figures.

Unified dose units are now explained in more detail.

5 injections have been performed. However, the clinical effect has been controlled of only 4 injections.

The upward error bars are much larger than the downward error bars. In Tab. 1 the standard deviation of the TSUI-score at BV0 is much larger than at AV0.

Reviewer 2 is right:

This paper does not recommend a ratio between abo- and rimaBoNT.

We have picked up the point, that 500 Us are recommended as initial dose for abo-treatment and 2500 - 10000 Us as initial dose for rima-treatment.

Therefore a 1:10 ratio is a good compromise.

We have modified our statement and use the SPCs.

Order of figures is changed.

We are sorry for that. Line 102 to 103 is the legend to Fig. 3. 

Results

Section 3.1              A table of patient demographics should be included for clarity.

Lines 121-122                    Conversely, does this mean that the side effects for BoNT/A treated patients declined over time because the dose was decreased?  If so, the authors should state that.

Lines 137-142                    The authors discuss the results for the single patient in weeks, but the data presented in the corresponding figure are in days.  This difference should be corrected.

Line 148 et seq                  The authors discuss significance levels here.  These should also be shown directly on the data in the figure.

Lines 173-176                    Are the authors referring to Figure 4 here?

Tab. 1 is included.

This is mentioned now.

We have now changed number in weeks into the  corresponding number of days.

Levels of significance are now indicated by symbols in the figure.

Reviewer 4 is right:

Fig. 4 is mentioned now.

Discussion

Line 181                  Cannot refer to Figure 1 (single patient data) here together with the other figures as these data in Figure 1 are for one injection of either toxin alone.

Lines 185-186                    The ONA: ABO ratio of 1:3 or 1:4 stated here is not as recommended in citation 23.  See below.

Lines 190-191                    This statement about the Pappert study cannot be made as that was a non-inferiority study: superiority cannot be determined from such a study.

Line 197 (also previously)            The authors should describe and illustrate the staircase effect they refer to

Line 201                  The term “highly negatively selected” is strange and should be revised

Lines 206-206                    The present authors have not presented appropriate data on non-inferiority (other than observational) and so this statement cannot be made.

Lines 220-221         A reduction in efficacy of about 50% cannot be described as “moderate”

Lines 235-237                    The authors have made a statement about the ratio of toxin target cleavage but presented no data on this molecular aspect.  This is incorrect and based on an assumption about the clinical doses they have used.  What are the ratios of SNAP25 and VAMP in human neurons?  Is the efficacy of toxin uptake the same for types A and B?  We know already that the binding efficacy of type B to human neurons is very low (STROTMEIER, J., WILLJES, G., BINZ, T. & RUMMEL, A. 2012. Human synaptotagmin-II is not a high affinity receptor for botulinum neurotoxin B and G: increased therapeutic dosage and immunogenicity. FEBS Lett, 586, 310-3). This must be taken into account in any comments about target cleavage. 

Lines 238-241         Need to be supported by citations on the various issues the authors raise or this part should be removed.

Fig. 1 is not mentioned anymore.

We did not write that 1:4 was recommended, but we have omitted 1:4 now.

We add that the tendency was non-significant.

This is done now.

We eliminate these terms and replace it by complex.

We have modified this sentence now.

Reviewer 2 is right:

We have omitted „moderate“.

We are thankful for these helpful comments. We have included most of them into the revised version of the manuscript.

We have removed this paragraph.

Reviewer 3 Report

  1. Study design:  Is this purely a retrospective study or was there a specific protocol followed or was this a study designed to prospectively follow as secondary non-responders?  Were all these patients receiving aboBoNT when they developed antibodies? Was the same provider performing the injections on these patients during the study period?  What criteria did the injector use to adjust dose of toxin injected int the study? Control visits were referred in the results section however not introduced in the study design. Please clarify how the study was organized in more detail including description of “control” visits.
  2. It is difficult to understand what data Figure 3 is showing.  What is the frequency represent and data from how many patients is analyzed in this and how many patients call into each category of the “frequency”.  I would suggest clarifying the title of the y-axis and writing in the number of patients each bar graph represents.
  3. Figure 2 needs some clarification.  Explanation of the figures in more detail is needed for the caption. They are first referenced earlier in the results section and then later described in more detail. Are the data points the average of of all 17 patients? This is very concentrated and given the sample size more detailed information needs to be provided about treatment response for each individual. It would be helpful to know when the repeat injections were given over the 400 days post-first injections and whether all patients received injections at 12 weeks or if there were some that received them earlier or later.
  4. In the section comparison of first injections in a single subject it is mentioned that 1000MU aboBoNT was used however it is then reported that 7500U rimBoNT was used. What injection units of aboBoNT were used? And what dilution ratio was used? Was it the same dilution ratio for all 17 patients? It would be helpful to know what the doses of aboBoNT were used on the 17 patients and what doses of rimBoNT were used on each of these patients.
  5. In Figure 1 what does “patient’s global assessment” represent. Does 100% represent the best effect of the medication (ie symptoms resolved) or does it represent back to baseline dystonia. Reviewing this it appears that the peak benefit should be around 45 days which correlate with an assessment percentage of ~65 and 75. Details on the values represent and how this assessment was obtained would be helpful to understand the data.
  6. “The personal score clearly demonstrates that the duration of clinical effect… may exceed 12 weeks” is difficult to interpret from the data presented.
  7. The discussion is lengthy the information presented is not relevant to the discussion of patients experiencing secondary treatment failure to toxin.

Author Response

Reviewer 3:

  1. Study design:  Is this purely a retrospective study or was there a specific protocol followed or was this a study designed to prospectively follow as secondary non-responders?  Were all these patients receiving aboBoNT when they developed antibodies? Was the same provider performing the injections on these patients during the study period?  What criteria did the injector use to adjust dose of toxin injected int the study? Control visits were referred in the results section however not introduced in the study design. Please clarify how the study was organized in more detail including description of “control” visits.

The materials and methods section is rewritten.

The study was purely retrospective.

All patients received aboBoNT/A only before switch to rimaBoNT/A.

Patients were treated by different physicians.

Visits are explained now

  1. It is difficult to understand what data Figure 3 is showing.  What is the frequency represent and data from how many patients is analyzed in this and how many patients call into each category of the “frequency”.  I would suggest clarifying the title of the y-axis and writing in the number of patients each bar graph represents.

The legend to Fig. 3 is extended now.

Frequency is equal to number of patients. This is explained in the legend now.

  1. Figure 2 needs some clarification.  Explanation of the figures in more detail is needed for the caption. They are first referenced earlier in the results section and then later described in more detail. Are the data points the average of of all 17 patients? This is very concentrated and given the sample size more detailed information needs to be provided about treatment response for each individual. It would be helpful to know when the repeat injections were given over the 400 days post-first injections and whether all patients received injections at 12 weeks or if there were some that received them earlier or later.

Figures are now reordered.

In Fig. 2 data points are averages of all 17 patients.

Treatment responses of all patients after two BoNT/A injections and after two BoNT/B injections is presented in Fig. 4.

Mean duration and standard deviation of all 4 controlled treatment cycles are presented in Tab. 2 now.

  1. In the section comparison of first injections in a single subject it is mentioned that 1000MU aboBoNT was used however it is then reported that 7500U rimBoNT was used. What injection units of aboBoNT were used? And what dilution ratio was used? Was it the same dilution ratio for all 17 patients? It would be helpful to know what the doses of aboBoNT were used on the 17 patients and what doses of rimBoNT were used on each of these patients.

Mean initial dose and standard deviation of aboBoNT/A and rimaBoNT/B are presented in Tab. 1 and Fig. 2B.

Dilution: aboBoNT/A: 500 U/2.5 ml

               rimaBoNT/B: 5000U/2.5 ml

Dilution ratio was the same for all patients.

  1. In Figure 1 what does “patient’s global assessment” represent. Does 100% represent the best effect of the medication (ie symptoms resolved) or does it represent back to baseline dystonia. Reviewing this it appears that the peak benefit should be around 45 days which correlate with an assessment percentage of ~65 and 75. Details on the values represent and how this assessment was obtained would be helpful to understand the data.

100% is severity at onset of BoNT/A therapy. Reduction of the assessment value means improvement reaching its peak effect after 45 days .

The legend to Fig. 1 and the corresponding text are improved.

  1. “The personal score clearly demonstrates that the duration of clinical effect… may exceed 12 weeks” is difficult to interpret from the data presented.

A line at day 84 (=12 weeks) helps to demonstrate that the clinical effect of had not declined down to the baseline level at day 84.

  1. The discussion is lengthy the information presented is not relevant to the discussion of patients experiencing secondary treatment failure to toxin.

Parts of the original discussion were omitted and a further paragraph 3.6 is added:

„Implications for patients with secondary treatment failure“.

Round 2

Reviewer 1 Report

Manuscript was improved and now is worthy for publication in Toxins journal.

Author Response

Thank you so much for your time and also nice and very helpful comments.

Reviewer 2 Report

The authors have carried out major revisions to their original submission.  These revisions are acceptable.

Author Response

(The authors gave the same response as above.)
